# The Spread of SARS-CoV-2 Variant Omicron with a Doubling Time of 2.0–3.3 Days Can Be Explained by Immune Evasion

**DOI:** 10.3390/v14020294

**Published:** 2022-01-30

**Authors:** Frederic Grabowski, Marek Kochańczyk, Tomasz Lipniacki

**Affiliations:** Institute of Fundamental Technological Research, Polish Academy of Sciences, 02-106 Warsaw, Poland; frederic.grabowski@ippt.pan.pl (F.G.); mkochan@ippt.pan.pl (M.K.)

**Keywords:** COVID-19 pandemic, SARS-CoV-2, Omicron variant, genome sequencing, mutation

## Abstract

Omicron, the novel highly mutated SARS-CoV-2 Variant of Concern (VOC, Pango lineage B.1.1.529) was first collected in early November 2021 in South Africa. By the end of November 2021, it had spread and approached fixation in South Africa, and had been detected on all continents. We analyzed the exponential growth of Omicron over four-week periods in the two most populated of South Africa’s provinces, Gauteng and KwaZulu-Natal, arriving at the doubling time estimates of, respectively, 3.3 days (95% CI: 3.2–3.4 days) and 2.7 days (95% CI: 2.3–3.3 days). Similar or even shorter doubling times were observed in other locations: Australia (3.0 days), New York State (2.5 days), UK (2.4 days), and Denmark (2.0 days). Log–linear regression suggests that the spread began in Gauteng around 11 October 2021; however, due to presumable stochasticity in the initial spread, this estimate can be inaccurate. Phylogenetics-based analysis indicates that the Omicron strain started to diverge between 6 October and 29 October 2021. We estimated that the weekly growth of the ratio of Omicron to Delta is in the range of 7.2–10.2, considerably higher than the growth of the ratio of Delta to Alpha (estimated to be in in the range of 2.5–4.2), and Alpha to pre-existing strains (estimated to be in the range of 1.8–2.7). High relative growth does not necessarily imply higher Omicron infectivity. A two-strain SEIR model suggests that the growth advantage of Omicron may stem from immune evasion, which permits this VOC to infect both recovered and fully vaccinated individuals. As we demonstrated within the model, immune evasion is more concerning than increased transmissibility, because it can facilitate larger epidemic outbreaks.

## 1. Introduction

Omicron, the novel SARS-CoV-2 Variant of Concern (VOC, Pango lineage B.1.1.529, Nextstrain clade identifier 21K) was first collected in South Africa on 2 November 2021, (GISAID [1] sequence accession ID: EPI_ISL_8182767). Compared to the original SARS-CoV-2 virus, Omicron carries 30 amino acid non-synonymous substitutions, three small deletions, and one small insertion in the spike protein [2]. Altogether, Omicron has 51 amino-acid level mutations, and its closest known sibling has 15 mutations (GISAID sequence accession ID: EPI_ISL_622806) with only nine common mutations, implying a distance of 42 mutations from the last common ancestor (based on the phylogenetic tree generated by Nextstrain [3]). The collection date of the sibling genome, 13 September 2020, suggests more than a year of evolution in an isolated niche, possibly in an immunocompromised host, but more data is necessary to rule out or confirm the existence of hidden branches (see [4] for discussion). The lineage started spreading rapidly in South Africa’s Gauteng province in November 2021, approaching fixation in the whole of South Africa by the end of that month and causing abrupt epidemic outbreaks across South Africa, then Europe, and finally other continents. In all these locations, Omicron outcompeted Delta VOC (lineage B.1.617.2), which in October 2021 accounted for more than 99% of genomes sequenced in Europe, North America, and Oceania, more than 90% in Asia and South America, and nearly 90% in Africa.

The rapid spread of Omicron both in South Africa, having a widespread infection-induced seroprevalence [5], and in Western European countries, where a high proportion of the population is vaccinated [6], suggests immune evasion that may be linked to the high number of mutations in viral spike glycoprotein, the major target of neutralizing antibodies [7]. This is confirmed by a growing number of (i) in vitro studies showing Omicron resistance to humoral immunity provided by vaccine- or pre-existing variant infection-induced antibodies, as well as epidemiological studies indicating (ii) significantly reduced vaccine effectiveness against infection with Omicron and (iii) higher chance of reinfections with Omicron compared to Delta.

(i)For two mRNA-based vaccines, BNT162b2 (Pfizer) and mRNA-1273 (Moderna), Liu et al. demonstrated a >21-fold and >8.6-fold decrease (Omicron versus D614G) in ID50 (infectious dose), respectively. For two vector vaccines, Ad26.COV2.S (Johnson & Johnson) and ChAdOx1 (AstraZeneca), all samples obtained from patients without a previous history of SARS-CoV-2 infection were below the level of detection against Omicron [8]. After three homologous mRNA vaccinations, the average ID50 drop was 6.5-fold [8]. Planas et al. showed that sera from either BNT162b2 or ChAdOx1 vaccine recipients (sampled 5 months after complete vaccination) barely inhibited Omicron. Sera from COVID-19 convalescent patients (collected 6 or 12 months post symptoms) displayed low-to-absent neutralizing activity against Omicron, whereas administration of a booster dose of BNT162b2 as well as vaccination of previously infected individuals generated an anti-Omicron neutralizing response, but with titers 5–31-fold lower than against Delta [9]. Omicron VOC was found to be 5.3–7.4-fold less sensitive than Beta VOC when assayed with serum samples obtained from individuals inoculated with 2 mRNA-1273 doses [10]. A meta-analysis of 24 studies showed a decrease in the neutralization titer (not significantly different between different vaccines) compared to the ancestral virus for the previous four VOCs: Alpha (1.6-fold), Gamma (3.5-fold), Delta (3.9-fold), and Beta (8.8-fold) [11]. This loss of neutralization activity is not as substantial as in the case of Omicron [8]. These findings are in line with another study that shows barely detectable serum neutralizing activity against Omicron after two mRNA vaccination doses (and still much lower neutralizing activity after the “booster” dose in relation to wild-type virus as well as the Delta VOC) [12].(ii)Andrews et al. showed a decrease in vaccine effectiveness against symptomatic infection by Omicron with respect to Delta [13]. Half a year after two-dose ChAdOx1 vaccination, the effectiveness was 42% against Delta, with no effect observed against Omicron starting 15 weeks after the second ChAdOx1 vaccination. In the case of BNT162b2, the protection 15 weeks after vaccination was 63% against Delta and 34–37% against Omicron. The BNT162b2 booster increases protection to above 93% against Delta and 75% against Omicron [13]. A report from the UK Health Security Agency confirms these results and additionally indicates that the mRNA “booster” effect against Omicron, but not against Delta, wanes rapidly in time to about 40% 10 weeks post “booster” dose [14]. These findings are in line with a report from the MRC Centre for Global Infectious Disease Analysis indicating a significantly increased risk of an Omicron case compared to Delta for those with vaccine status AZ 2+weeks post Dose 2 (PD2), Pfizer 2+w PD2, AZ 2+w post Dose 3 (PD3) and PF 2+w PD3 vaccine states with hazard ratios of 1.86 (95% CI: 1.67–2.08), 2.68 (95% CI: 2.54–2.83), 4.32 (95% CI: 3.84–4.85), and 4.07 (95% CI: 3.66–4.51), respectively [15].(iii)The same report indicates that Omicron is associated with a 5.41 (95% CI: 4.87–6.00)-fold higher risk of reinfection compared with Delta [15].

Although all the evidence is based on a limited number of cases and may be influenced by population-level biases, one can expect that the hazard ratio of Omicron versus Delta infection is in the range 2–5 and depends principally on specific vaccines, the proportion of the population vaccinated by the “booster” dose, and resistance after recovery from COVID-19.

In this work, we estimated the Omicron variant doubling time in South Africa based on the daily number of new COVID-19 cases in the two most populated South African provinces, Gauteng and KwaZulu-Natal [16]. Further, we repeated this estimation for Australia, the UK, Denmark, and New York State, four locations that differ with respect to their COVID-19 epidemic histories and proportions of COVID-19 vaccine manufacturers. In all these locations, new COVID-19 cases are expected to be relatively reliably reported and to have their viral genomes screened by sequencing. Based on data for the three latter locations, we showed that the weekly growth of the ratio of Omicron to Delta cases significantly exceeds previous weekly growths of the ratio of Delta to Alpha cases, and the ratio of Alpha cases to cases caused by pre-existing strains. To rule out hidden spread of the Omicron variant, which could potentially influence the above estimates, we used an Omicron phylogenetic tree from Nextrain to estimate the strain divergence date. Finally, we employed a two-strain mathematical model to demonstrate that observed rapid outbreaks of Omicron strain can be explained solely by immune evasion, which expands the pool of individuals susceptible to infection.

## 2. Materials and Methods

### 2.1. Genomic Sequence-Based Analysis

All data used in this study were retrieved as of 9 January 2022. The GISAID [1] data include genomes with submission dates earlier than 6 January 2022.

In Figure 1 the weekly cases of Omicron, Delta, and pre-existing strains were estimated based on GISAID data and the cumulative number of COVID-19 cases in Gauteng and KwaZulu-Natal (DSFSI at the University of Pretoria, https://github.com/dsfsi/covid19za/tree/master/data, accessed on 9 January 2022). For Gauteng this dataset was amended by changing the number of cases from 8099 to 605 for 23 November 2021 using information from the South African National Institute for Communicable Diseases (NICD, https://www.nicd.ac.za/latest-confirmed-cases-of-covid-19-in-south-africa-23-november-2021, accessed on 9 January 2022). According to NICD, the difference results from a retrospective addition of 7494 antigen test results. The dataset used for Figure 1A,C is provided as Appendix A. 

Mobility in Gauteng and KwaZulu-Natal (Figure 1B) was assessed based on the COVID-19 Community Mobility Reports from Google (https://www.google.com/covid19/mobility, accessed on 9 January 2022). Weekly averages were calculated based on workdays.

The Omicron strain divergence date and mutation accumulation rate (Figure 1D) were determined by Poisson regression, assuming that the mean number of mutations grows linearly with time. The Omicron phylogenetic tree (with last update on 3 January 2022) was retrieved from Nextstrain [4]; sample collection dates of genome sequences selected by Nextstrain were retrieved from GISAID. The resulting dataset is provided as Appendix A.

Two lists of GISAID IDs and corresponding acknowledgments for all genomes collected from 19 April 2021 until 2 January 2022 in Gauteng and KwaZulu-Natal are provided as Appendix A.

In Figure 2 and Figure 3 the weekly cases of Omicron, Delta, and pre-existing strains were estimated based on sequence data from GISAID and case data aggregated by Johns Hopkins University [17]. The datasets used for Figure 2 and Figure 3 are provided as Appendix A.

The Omicron doubling time (Figure 1C, Figure 2B,E and Figure 3B,E) was estimated by log–linear regression of the (estimated) number of Omicron cases in the exponential phase of its growth. Daily new cases were aggregated by week. An appropriate four-week period was used in each location except Australia, for which a six-week period was used.

The weekly (multiplicative) growth rates of the ratios of an emerging to a pre-existing strain (Figure 2C,F and Figure 3C,F) were also estimated by log–linear regression. To enable comparison between emerging strains, we select four-week periods of the fastest growth, so that in at least two of those weeks the emerging strain has lower counts than the pre-existing one. This additional criterion ensures that the analyzed period captures the emergence of the new strain. We use the reporting convention in which the weekly (multiplicative) growth rate of two implies doubling of the strain-to-strain ratio every week.

### 2.2. Mathematical Modeling

The mathematical model has been formulated as an extended susceptible–exposed–infectious–recovered (SEIR) model amended with a vaccinated (V) compartment. We assumed that the latent period was the same as the incubation period and was Erlang-distributed with the shape parameter *m* = 6 (which in the model structure is reflected by the inclusion of six exposed subcompartments) and the mean of 1/σ = 3 days. The average period of infectiousness is 1/γ = 3 days (such a short period reflects the assumption that the individuals with confirmed infection are quickly isolated and then cannot infect susceptible individuals). The recovered individuals become susceptible at the rate of ρ = 1/year. Susceptible individuals are vaccinated at the rate of ν = 2/year, and their vaccine-induced immunity wanes at the rate ρ (same as the rate of transition from a recovered to a susceptible compartment).

We consider two model variants—see Figure 4A,B. In Model A, there is a single pool of individuals susceptible to both Delta and Omicron, whereas in Model B there are two additional compartments of individuals that are susceptible to either only Delta or only Omicron, and that are fed with the post-Omicron recovered or the post-Delta recovered, respectively, at the rate of π = 1.5/year. From these two additional susceptible compartments there are transitions to the compartment of individuals susceptible to both Delta and Omicron (at the rate ρ). In this way we account for only partial overlap in reciprocal post-infection protection. In Model A, transmissibility of Omicron is 4-fold higher than that of Delta, whereas in Model B the transmissibility of Omicron and Delta is the same but, due to the specific choice of transition parameters, the aggregated pool of individuals susceptible to Omicron is four-fold higher than that of individuals susceptible to Delta in the steady state before the appearance of Omicron. Model B is symmetrical with respect to both strains; however, by including an additional transition from the vaccinated compartment to the compartment of individuals susceptible only to Omicron (at the rate π), we account for faster waning of post-vaccination immunity to Omicron.

Model dynamics are governed by a system of ordinary differential equations (18 ODEs in Model A, 20 ODEs in Model B). The ODEs may be unambiguously derived, assuming mass-action kinetics, based on model schemes in Figure 4A,B, and parametrized with kinetic rates given in Figure 4C. On the first day shown in Figure 4D–G, when the system is in equilibrium with Delta, just one individual (in a population of 10^6^ individuals) is exposed to Omicron. As emphasized further in Results, the initial exponential phase of the Omicron outbreak is nearly identical within both models.

## 3. Results

### 3.1. Divergence and Growth of Omicron Strain in South Africa

The Delta VOC became the dominant variant in Gauteng and KwaZulu-Natal in June 2021, causing an epidemic wave that peaked at the beginning of July 2021 in Gauteng and the end of August 2021 in KwaZulu-Natal, Figure 1A. The number of Omicron (or other variant) cases were estimated by multiplying the weekly number of total COVID-19 confirmed cases by the proportion of Omicron (or other variant) genomes among all collected genomes in a given week (see Appendix A). Between July and October 2021 in Gauteng and in September and October in KwaZulu-Natal, the weekly number of COVID-19 cases was decreasing despite no significant reduction in population mobility at workplaces and retail and recreation centers (Figure 1B). The emergence of the Omicron variant has caused recent rapid epidemic outbreaks in both provinces considered (Figure 1A). In Figure 1C we show the exponential growth of Omicron variant cases in weeks 45–48 in 2021 (8 November–5 December) in Gauteng and in weeks 46–49 in Kwazulu-Natal. This method, in contrast to analyzing only the proportion of new strain genomes [18], enabled us to follow Omicron growth after its fixation in weeks 47–48 in 2021. The Omicron doubling time, estimated based on the log–linear regression of the number of weekly cases in the four-week periods, is equal to 3.3 days (95% CI: 3.2–3.4 days) in Gauteng, and 2.7 days (95% CI: 2.7–3.3 days) in KwaZulu-Natal.

The log–linear regression suggests that the exponential growth of Omicron started in Gauteng around October 11, 2021; however, the initial epidemic growth is highly stochastic and may be heavily disturbed by superspreaders in the cascade of infections [19]. The profile of mutation accumulation in Figure 1D indicates that the Omicron strain started diverging between 6 October and 29 October 2021 (95% CrI), at an average mutation accumulation rate equal 0.33/week (95% CrI: 0.26–0.40 per week). This is lower than the average (global) SARS-CoV-2 mutation accumulation rate equal to approximately 0.45/week (based on the Nextstrain [3] estimate as of 15 January 2021 [20]). An assumption of a higher mutation rate would yield a later divergence date.

### 3.2. Succession of SARS-CoV-2 Variants of Concern

In Figure 2 we analyze dynamics of Omicron-driven COVID-19 outbreaks in the UK and Denmark, and compare them with outbreaks caused therein by two previous VOCs, Alpha and Delta. In both these countries, the Alpha variant outcompeted pre-existing strains, approaching fixation, and then it was outcompeted by the Delta variant, which also approached fixation; in December 2021, Omicron became the dominant strain (Figure 2A,D). We estimate the doubling time of Omicron to be 2.4 and 2.0 days in the UK and in Denmark, respectively, based on its nearly exponential growth in the 4-week period from 22 November to 19 December 2021, Figure 2B,E. The short doubling time of Omicron is on par with its rapid gain of dominance over the Delta VOC. In the 4-week period considered, the Omicron:Delta ratio increases exponentially with a weekly growth rate of 8.1 in the UK and 10.2 in Denmark. This is faster than the earlier growth of the Delta:Alpha ratio, estimated to be 3.2 in the UK and 4.2 in Denmark, and much faster than the growth of Alpha with respect to pre-existing strains, which was 2.7 in the UK and 2.0 in Denmark. All data used in Figure 2 are provided in Appendix A.

In Figure 3 we analogously analyze Omicron-driven outbreaks in New York State and Australia, and again compare them with outbreaks caused by two previous VOCs, Alpha and Delta. In New York State, the Alpha variant arrived during the end phase of the second wave, and consequently caused a noticeable but relatively modest rise in cases. It was then outcompeted by Delta, which, causing the third epidemic wave, quickly reached fixation. In Australia, due to stringent lockdowns and strict border rules, there were relatively few COVID-19 cases before the Delta variant. In both regions, Omicron became the dominant variant within one month after its first detection (Figure 2A,D). We estimate the doubling time of Omicron to be 2.5 and 3.0 days in New York State and Australia, respectively (Figure 2B,E). In the 4-week period considered, the Omicron/Delta ratio was found to grow exponentially with a weekly rate 7.7 in New York State and 7.2 in Australia. This is significantly higher than the Delta/Alpha and Alpha/pre-existing strains growth rates in the New York State, estimated to be 2.5 and 1.8, respectively. All data used in Figure 3 are provided in Appendix A.

### 3.3. Two-Strain Mathematical Model

To corroborate whether the observed rapid surge of Omicron cases that displace Delta may be attributed to immune evasion, as widely suggested by the references cited in the Introduction, we analyzed two variants of a mathematical model of the COVID-19 pandemic (Figure 4). In Model A (Figure 4A), transmissibility of Omicron is four-fold higher than that of Delta and both strains share a common pool of susceptible individuals. In Model B (Figure 4B), transmissibility of Omicron and Delta is identical but, in the steady state before the appearance of Omicron, the aggregated pool of individuals susceptible to Omicron is four-fold higher than that of individuals susceptible to Delta. The models were structured and parametrized (Figure 4C) such that the doubling time of Omicron cases in both models is equal to 2.5 days (Figure 4D), which lies roughly in the middle of the range of doubling times observed in six geographical locations considered previously. Moreover, the ratio of new Omicron to new Delta cases is accordant in both models and equal to 6.9 days (Figure 4E), only slightly lower than in the four locations for which this ratio was determined.

Despite having non-distinguishable initial exponential phases, in later time points the two models exhibit divergent trajectories that differ markedly in terms of the peak of the number of daily new Omicron cases. According to the model, with increased Omicron transmissibility (Model A), the maximum number of new daily cases is below 1% of the population (Figure 4F). In the model with the increased pool of individuals susceptible to Omicron (Model B), the initial exponential growth phase lasts longer, which contributes to delaying and, most importantly, elevating this maximum to above 3% of population (Figure 4G).

## 4. Discussion

We have demonstrated the exponential growth of the Omicron strain in the South African provinces of Gauteng and KwaZulu-Natal in the four-weeks period starting, respectively, on November 8 and 15, 2021, with the doubling times equal to 3.3 days (95% CI: 3.2–3.4 days) and 2.7 days (95% CI: 2.3–3.3 days). Based on the mutation accumulation profile, we found that the Omicron strain started diverging between 6 October and 29 October 2021, which agrees with the date suggested by the log–linear regression of the number of weekly cases in the first affected province of Gauteng, 11 October 2021. Notably, an unnoticed spread before October 2021 would imply that the strain growth rate is lower than that estimated based on the exponential growth rate in the analyzed four-week period in Gauteng. Before the Omicron outbreaks, the Delta variant was the dominant strain in Gauteng and KwaZulu-Natal, and in September and October the COVID-19 epidemic was receding without significant mobility reduction, suggesting that the population of these provinces might have reached a transient herd immunity to the Delta variant. The population-level immunity has been apparently overcome by the Omicron variant.

The potential of Omicron to create rapid outbreaks was confirmed by analyzing its spread in the UK, Denmark, New York State, and Australia. In these locations the doubling time of Omicron was in the range of 2.0–3.0 days. Additionally, based on the relatively high number of sequenced genomes sampled in these locations, we estimated the weekly growth of the ratio of Omicron to Delta to be in the range of 7.2–10.2, considerably higher than the growth of the ratio of Delta to Alpha (estimated to be in in the range of 2.5–4.2), and Alpha to pre-existing strains (estimated to be in the range of 1.8–2.7). These findings are in line with the observed ability of Omicron to infect vaccinated and recovered individuals, which endows it with a natural advantage over Delta [15]. Notably, the Alpha outbreak took place in winter 2020/2021, when the proportion of vaccinated individuals was very low. In turn, the reduction of vaccine effectiveness for Delta in reference to Alpha [11,21] was substantially lower than in the case of Omicron in relation to Delta [15]. This suggests that, in contrast to Omicron, the Alpha and Delta variants become transiently dominant mainly because of their higher infectivity and not due to significant immune evasion.

In the six locations considered, the Omicron doubling time was found to be in the range of 2.0–3.3 days, which is comparable to the doubling times during the first COVID-19 pandemic outbreaks in spring 2020. For that time, doubling time was estimated to lie between 1.86 and 2.88 days for China, Italy, France, Germany, Spain, UK, Switzerland, and New York State [19]. Here, the weekly growth of the ratio of Omicron to Delta was found in the range of 7.2–10.2, considerably higher than the previous ratios of Alpha and Delta at the times when they were gaining dominance. These findings strongly suggest that Omicron will outcompete Delta and become (transiently) the dominant strain.

Omicron accumulated more than 30 mutations in its spike protein, with 15 substitutions in the receptor binding domain (RBD, residues 319–541) alone [2]. Many of these RBD mutations are thought to decrease potency of neutralizing antibodies [22], which is in agreement with growing evidence that Omicron has a several-fold increased ability to infect both vaccinated and recovered individuals, as discussed in the Introduction.

Our mathematical model-based analysis of COVID-19 dynamics clearly demonstrates two points relevant to the initial phase of the outbreak. First, if only rough epidemiological data are analyzed, in the first weeks of the outbreak immune evasion may be indistinguishable from increased transmissibility. Second, being able to distinguish between the two scenarios based on tangential evidence supporting immune evasion is of crucial importance for predicting the impact of a new strain on longer term epidemic dynamics. Immune evasion is more concerning than increased transmissibility, because dodging protection provided by vaccination or infection with prior variant(s) renders a significant share of the population susceptible to an emerging variant, promoting larger outbreaks. These outbreaks may be hard to curb by lockdowns due to increasing lockdown fatigue, but in the case of Omicron plausibly will not result in proportionally high death toll, as suggested by early estimates of Omicron-associated mortality [5,23].

## Figures and Tables

**Figure 1 viruses-14-00294-f001:**
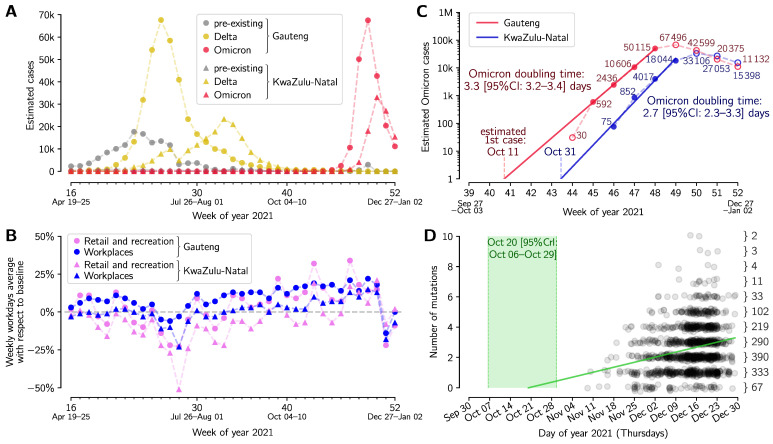
Growth and divergence of the Omicron strain. (**A**) Weekly aggregated cases of Omicron, Delta and other variants in two South African provinces, Gauteng and KwaZulu-Natal. (**B**) Weekly averaged workday mobility in Gauteng (filled circles) and KwaZulu-Natal (filled triangles) in workplaces (blue) and retail and recreation centers (pink). (**C**) Exponential growth of the Omicron strain in weeks 45–48 in 2021 (8 November–5 December) in Gauteng and in weeks 46–49 in KwaZulu-Natal. (**D**) Accumulation of mutations by the Omicron strain worldwide based on the Nextstrain phylogenetic tree. The green line shows the mutation accumulation trend determined by the linear regression assuming Poisson distribution of the number of mutations at a given time. The 95% credible interval of time is marked in light green. The dataset for panels (**A**,**B**) is provided as Appendix A; the phylogenetic tree (with dates) is provided as Appendix A.

**Figure 2 viruses-14-00294-f002:**
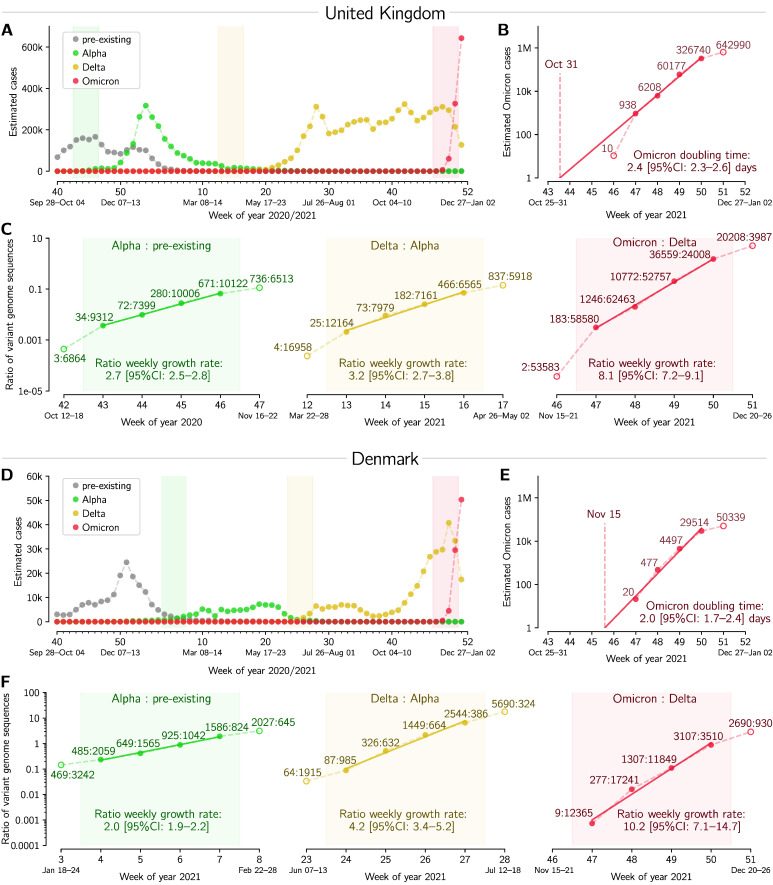
Succession of SARS-CoV-2 strains in the UK and in Denmark. (**A**,**D**) Estimated number of weekly cases infected with a particular strain over the weeks of 2020 and 2021. (**B**,**E**) Estimated number of weekly Omicron cases and the doubling time estimate based on log–linear regression in four whole-week periods (filled circles). (**C**,**F**) Ratios of weekly cases of an emergent strain to the previously dominant strain, and the estimate of ratios’ growth rates based on log–linear regression in four whole-week periods (subpanels correspond to shaded regions in respective panels (**A**,**D**)). A dataset for this figure is provided as Appendix A.

**Figure 3 viruses-14-00294-f003:**
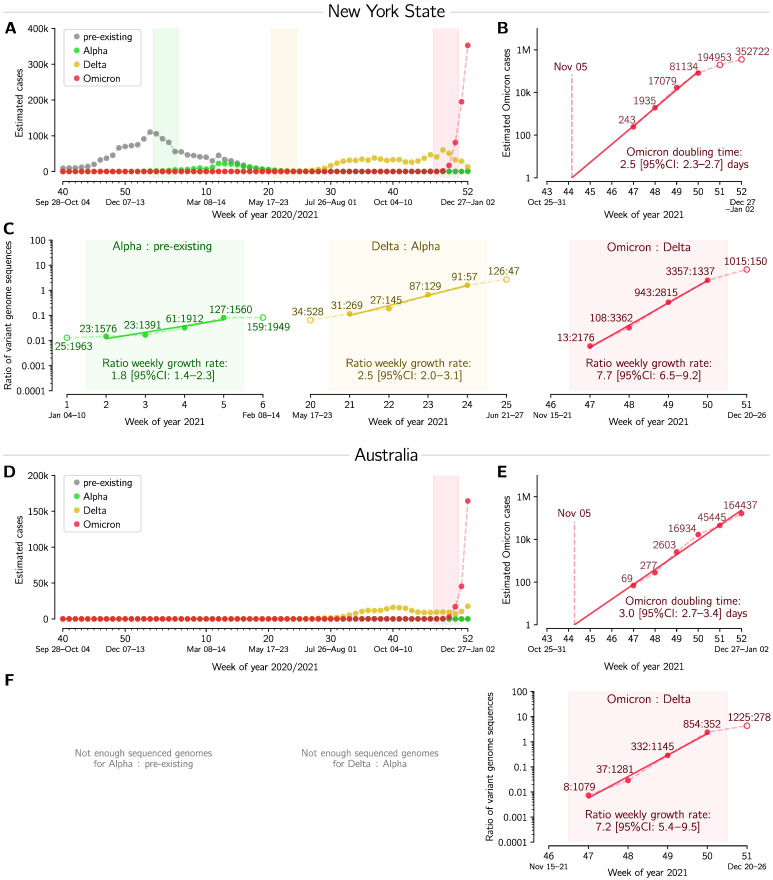
Succession of SARS-CoV-2 strains in New York State and in Australia. (**A**,**D**) Estimated number of weekly cases infected with a particular strain over the weeks of 2020 and 2021. (**B**,**E**) Estimated number of weekly Omicron cases and the doubling time estimate based on log–linear regression in four (Panel (**B**)) and six (Panel (**E**)) whole-week periods (filled circles). (**C**,**F**) Ratios of weekly cases of an emergent strain to the previously dominant strain, and the estimate of ratios’ growth rates based on log–linear regression in four whole-week periods (subpanels correspond to shaded regions in respective panels(**A**,**D**)). A dataset for this figure is provided as Appendix A.

**Figure 4 viruses-14-00294-f004:**
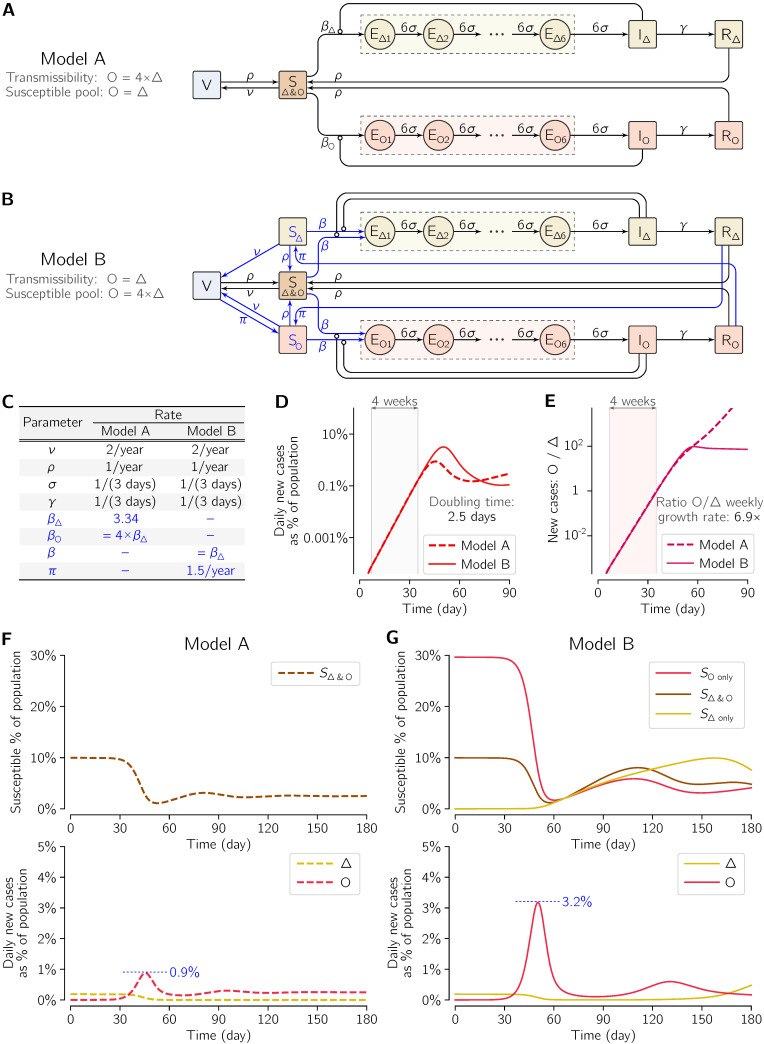
Two COVID-19 two-strain SEIR models with vaccination. (**A**) Scheme of Model A, in which the transmissibility of Omicron (O) is four-fold higher than that of Delta (Δ) and both strains have a common pool of susceptible individuals (10% of simulated population in the pre-Omicron steady state). (**B**) Scheme of Model B, in which the transmissibility of Omicron (O) and Delta (Δ) is the same but the aggregated pool of individuals susceptible to Omicron is four-fold higher than to Delta (40% vs. 10% of simulated population in the pre-Omicron steady state). Essential modifications with respect to Model A are shown in blue. (**C**) Values of rate parameters of both models. Model variant-specific parameters are blue. (**D**) Initial dynamics of Omicron dynamics in both models shows similar growth and identical doubling time in the 4-week time window in the initial exponential phase of the epidemic outbreak, but not in later time points. (**E**) Ratio of Omicron to Delta new daily cases and its growth rate in both models. (**F**) Dynamics of the outbreak of Omicron infections in Model A (**G**).

## Data Availability

We analyzed publicly available genome datasets retrieved from GISAID (https://www.gisaid.org, accessed on 9 January 2022) and Community Mobility Reports available from Google (https://www.google.com/covid19/mobility/, accessed on 9 January 2022). Cases data for Gauteng and KwaZulu-Natal can be found in daily reports from the National Institute for Communicable Diseases (https://www.nicd.ac.za, accessed on 9 January 2022) as aggregated by the University of Pretoria (https://github.com/dsfsi/covid19za, accessed on 9 January 2022). Cases for all other regions can be found in data aggregated by Johns Hopkins University (https://github.com/CSSEGISandData/COVID-19, accessed on 9 January 2022). Mutation counts for Omicron sequences can be found on Nextstrain (https://nextstrain.org/groups/neherlab/ncov/21K-diversity, accessed on 9 January 2022).

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
