# Peer review of "The Spread of SARS-CoV-2 Variant Omicron with a Doubling Time of 2.0–3.3 Days Can Be Explained by Immune Evasion"

_viruses, 2022, doi:10.3390/v14020294_

Round 1

Reviewer 1 Report

This is a very interesting work on the origin and characteristics of the omicron variant.

However, some clarifications and corrections are needed.

First, In the title and twice in the manuscript, authors conclude that herd immunity to the delta variant has been achieved in Guateng province. The conclusion is based on the fact that the number of delta cases has been steadily decreasing since the peak in late June in the absence of significant mobility reduction. However, one can not conclude that herd immunity has been achieved only on the basis of decreasing numbers of cases. There are other possible reasons which can lead to the decrease of cases, e.g. seasonality, climate factors, of unknown/unidentified factors, non-pharmaceutical interventions in the community which are not related to mobility.

Secondly, Figure 1A, 1B and 1C show estimates for which there is no explanation how the estimates are achieved.

It should be explained in the text how estimated total cases per week, the estimated omicron cases per week and the weekly workdays average have been calculated.

Attached is the PDF of the manuscript with comments.

Bes regards

Reviewer 2 Report

The manuscript “Omicron strain spreads with the doubling time of 3.2–3.6 days in South Africa’s province of Gauteng, wherein herd immunity to Delta variant has been achieved” is addressing the current hot topic of novel SARS-CoV-2 variant of concern – Omicron’s origin and transmissibility. The results based on estimated number of Omicron cases and phylogenetic analysis are interesting, confirming the suspicions of extremely high growth rate and an abrupt outbreak of the new variant.

Author Response

We are grateful for positive assessment of our study

Round 2

Reviewer 1 Report

The authors have addressed my concerns appropriately, I'm OK with the manuscript now.